# Zebrafish Ski7 tunes RNA levels during the oocyte-to-embryo transition

**Luis Enrique Cabrera-Quio[1], Alexander Schleiffer[1], Karl Mechtler[2], Andrea Pauli[1]***

**1** Research Institute of Molecular Pathology (IMP), Vienna Biocenter (VBC), Vienna, Austria, **2** Institute of Molecular Biotechnology, IMBA, Campus Vienna-Biocenter (VBC), Vienna, Austria

* andrea.pauli@imp.ac.at

**Data Availability Statement:** Sequencing data has been submitted to GEO (GSE147112): https://www.ncbi.nlm.nih.gov/geo/query/acc.cgi?acc=GSE147112 Underlying numerical data for all graphs and summary statistics are provided as

## Abstract

Post-transcriptional regulation of gene expression is crucial during the oocyte-to-embryo transition, a highly dynamic process characterized by the absence of nuclear transcription. Thus, changes to the RNA content are solely dependent on RNA degradation. Although several mechanisms that promote RNA decay during embryogenesis have been identified, it remains unclear which machineries contribute to remodeling the maternal transcriptome. Here, we focused on the degradation factor Ski7 in zebrafish. Homozygous *ski7* mutant fish had higher proportions of both poor quality eggs and eggs that were unable to develop beyond the one-cell stage. Consistent with the idea that Ski7 participates in remodeling the maternal RNA content, transcriptome profiling identified hundreds of misregulated mRNAs in the absence of Ski7. Furthermore, upregulated genes were generally lowly expressed in wild type, suggesting that Ski7 maintains low transcript levels for this subset of genes. Finally, GO enrichment and proteomic analyses of misregulated factors implicated Ski7 in the regulation of redox processes. This was confirmed experimentally by an increased resistance of *ski7* mutant embryos to reductive stress. Our results provide first insights into the physiological role of vertebrate Ski7 as a post-transcriptional regulator during the oocyte-to-embryo transition.

## Author summary

During early embryo development, all cellular processes are regulated by maternal products deposited in the egg. One of such products is RNA, which serves as a template for the synthesis of proteins. As the embryo grows in the number of cells and size, RNAs need to be removed. Although many components and pathways that regulate RNA degradation in the embryo have been identified, the complete set of mechanisms are still unclear. Here, we characterize the function of the superkiller 7 (Ski7) protein, known to be involved in RNA degradation in yeast. We take advantage of high-throughput technologies to profile the status of RNAs during the transition from the egg to the embryo in zebrafish in the presence and abscense of Ski7. Absence of Ski7 in zebrafish leads to misregulation of hundreds of transcripts, decreases egg quality, and causes presumptive subfertility in female

Supplementary Tables. Code Availability Scripts generated for the processing and analyses of data in this manuscript are available on GitHub (https://github.com/Quio-Enrique/pauliLab_Ski7_OET).

**Funding:** L.E.C.Q. was supported by a Boehringer Ingelheim Fonds (BIF) PhD fellowship (https://www.bifonds.de/fellowships-grants/phd-fellowships.html). Work in A.P.'s lab has been supported by the FWF START program (Y 1031-B28), the HFSP Career Development Award (CDA00066/2015), the SFB RNA-Deco (project number F 80), and EMBO-YIP funds. K.M. has been supported by EPIC-XS, project number 823839, funded by the Horizon 2020 program of the European Union, and the Austrian Science Fund by ERA-CAPS I 3686 International Project. The IMP receives institutional funding from Boehringer Ingelheim and the Austrian Research Promotion Agency (Headquarter grant FFG-852936). The funders had no role in study design, data collection and analysis, decision to publish, or preparation of the manuscript.

**Competing interests:** The authors have declared that no competing interests exist.

fish. Overall, Ski7 is a previously unrecognized factor in vertebrates that is important for the regulation of RNA degradation during the transition from the egg to the embryo.

## Introduction

Post-transcriptional regulation of gene expression plays a major role in defining the amount of RNA and protein in a cell. Apart from ensuring normal cell physiology and maintaining cell fate, post-transcriptional control is important for the speed and robustness of cell fate changes during developmental transitions [1,2].

A crucial aspect of post-transcriptional control is RNA decay, which removes unneeded or aberrant RNAs. In eukaryotes, canonical RNA degradation initiates with the shortening of the polyA-tail [3–5]. Following deadenylation, RNAs are degraded by the 5'-to-3' and/or 3'-to-5' decay pathways [6]. The 5'-to-3' degradation pathway is initiated by the removal of the 5' cap by the Dcp1/Dcp2 decapping complex, followed by transcript elimination by the 5'-to-3' exonuclease Xrn1 [7,8]. Alternatively, deadenylated RNAs are degraded in a 3'-to-5' manner by the RNA exosome complex and several auxiliary cofactors [9,10].

The RNA exosome is an essential and highly conserved multiprotein complex composed of ten core factors. The catalytically inactive subunits of the exosome (members of the RNAse PH protein family) form a hexameric ring, which is capped by a trimeric ring of S1/KH domain-containing proteins (Csl4, Rrp40, Rrp4) [10,11]. This catalytically inactive nine-subunit complex associates with one of three exoribonucleases, Rrp6/EXOSC10, Rrp44/DIS3, or DIS3L in the nucleolus, nucleus, or cytosol, respectively [12]. Besides its distinct, subcellularly restricted nucleases, the exosome binds auxiliary factors in a compartment-specific manner. In the nucleus, the exosome is activated by the RNA helicase Mtr4 [13]. In contrast, the cytosolic exosome is stimulated by the Ski complex [10,14].

The Ski complex is an evolutionary conserved protein complex that participates in canonical 3'-to-5' RNA decay [15–17], RNA surveillance [17,18], viral RNA defense [19,20], and RNA interference [21]. Consistent with facilitating cytosolic exosome-dependent 3'-to-5' RNA degradation, studies in yeast have shown that mutants in Ski complex subunits are synthetic lethal with mutants in the 5'-to-3' degradation pathway [15,22]. However, the Ski complex, which contains the DExH-box helicase Ski2/SKIV2L, the tetratricopeptide repeat scaffold protein Ski3/TTC37, and two copies of the WD40 repeat protein Ski8/WDR61 [10], does not directly interact with the exosome. Association with the exosome complex requires the adaptor protein Ski7, which stably binds to the Csl4 exosome subunit via its conserved Patch 4 (P4)-like motif (ENFxxxSPDDIIQxAQ in yeast, PFDFxxxSPDDIVKxNQ in humans), and transiently associates with the Ski complex via the Ski3-Ski8 subunits [14,23–25]. In yeast Ski7, the exosome and Ski-complex interacting domains are part of the N-terminus [14,24], while the C-terminus consists of a GTPase-like domain involved in RNA quality control (non-stop decay pathway (NSD)) [18,26,27].

Yeast Ski7 is a paralogue of Hbs1, a factor required to degrade transcripts containing stalled ribosomes as part of the no-go decay (NGD) pathway [26,28,29]. While both yeast paralogues are encoded in independent genetic loci [30,31], the recently identified Ski7 homolog in vertebrates is encoded by an alternative splice isoform of vertebrate *hbs1l*, hence its name in humans HBS1LV3 [24,25,31]. Vertebrate Ski7 shares the Ski and exosome complex interacting domains of yeast Ski7, yet it lacks the GTPase domain [27,31,32]. Although a handful of studies have started to characterize vertebrate Ski7's function *in vitro* [24,25], Ski7's function *in vivo* has not yet been investigated in higher eukaryotes.

Here, we report a functional analysis of vertebrate Ski7 in zebrafish. We found that *ski7* mRNA levels peak during the oocyte-to-embryo transition. Regulation of this transition is fully dependent on post-transcriptional mechanisms including RNA degradation since the mature egg and early zebrafish embryo are transcriptionally silent [1,2]. Due to Ski7's implication in RNA degradation and its specific timing of expression, we investigated Ski7's role during the oocyte-to-embryo transition.

## Results

### Ski7 mRNA is highly expressed during the oocyte-to-embryo transition

Analysis of zebrafish transcriptome [33,34] and translatome [35,36] data revealed an *hbs1l* splice variant that is highly expressed and translated during early stages of embryogenesis (Fig 1A and 1B). Conservation analysis of this splice isoform showed that it encodes the zebrafish homolog of yeast Ski7 and human HBS1LV3 (Fig 1C and S1 Table), which is consistent with recent data from other eukaryotes [24,25,31]. The zebrafish *ski7*-specific exon (exon 5) contains the previously annotated Ski7-like motif and the most conserved region of Ski7, the so-called Patch 4-like (P4-like) motif [24,32] (Fig 1C). Analysis of *ski7* mRNA levels during oogenesis, in mature eggs, and during embryogenesis showed that *ski7* mRNA peaks during the oocyte-to-embryo transition: while expression of *hbs1l* remains constant, we found that *ski7* mRNA is >10-fold higher expressed than *hbs1l* in mature eggs, but 2-fold lower at the beginning of oogenesis and in 5-day old larvae (Fig 1B). A similar trend was observed in mouse (S1A Fig). Our finding that *ski7* mRNA is enriched in the mature egg raised the possibility that it may be important during the oocyte-to-embryo transition, which is characterized by large-scale changes to RNA and protein content.

### *Ski7* mutants produce eggs of poor quality and fewer embryos that develop

To analyze the function of Ski7 *in vivo* in a vertebrate organism, we generated zebrafish lacking full-length Ski7 protein. To not interfere with Hbs1l expression, we targeted the beginning of the *ski7*-specific exon 5 using CRISPR/Cas9 (Figs 1A and S1C). We obtained mutant fish carrying a 203-nucleotide deletion (S1B and S1C Fig) resulting in a frameshift mutation after leucine 183 (full-length Ski7: 576 amino acids). This allele is predicted to generate a truncated Ski7 protein lacking both conserved Ski7-like and P4-like motifs (S1B Fig), while *hbs1l* mRNA levels were unaffected (S1C Fig). This mutant will hereafter be referred to as *ski7*[-/-]. *Ski7*[-/-] adult fish were viable and lacked any overt morphological abnormalities (S2 Fig). However, we noticed that *ski7*[-/-] mutant females tend to lay eggs of poor quality (opaque, deformed) (S3A Fig and S2 Table). In addition, a significant fraction of good quality eggs did not develop past the one-cell stage. This suggests that even eggs that looked morphologically normal at the time of laying had either already defects during oogenesis or were defective at the stage of fertilization or early development (Kruskal-Wallis with Dunn's multiple comparisons test, p = $4.7e^{-06}$) (Fig 1D and S3 Table). Reciprocal crosses of *ski7*[-/-] mutant fish with wild-type fish showed a more pronounced defect if Ski7 was lacking in the female as opposed to the male (66.13% and 81.3% average percentage of developing embryos, respectively) (S3B Fig and S3 Table), which suggests that Ski7 is mainly required from the side of the female. Notably, the poor egg quality and reduced number of developing embryos were rescued by expression of transgenic GFP-tagged Ski7 protein (Fig 1D and S3 Table). This confirms that the defects observed in the *ski7*[-/-] fish were due to the lack of Ski7. However, *ski7*[-/-] embryos that successfully underwent the first cell cleavages did not display any morphological defects or developmental delays during embryogenesis (Fig 1E and 1F and S4 Table) and developed into morphologically normal adult fish (S2 Fig). The poor egg quality and reduced number of developing embryos of *ski7*[-/-] mutants combined with the

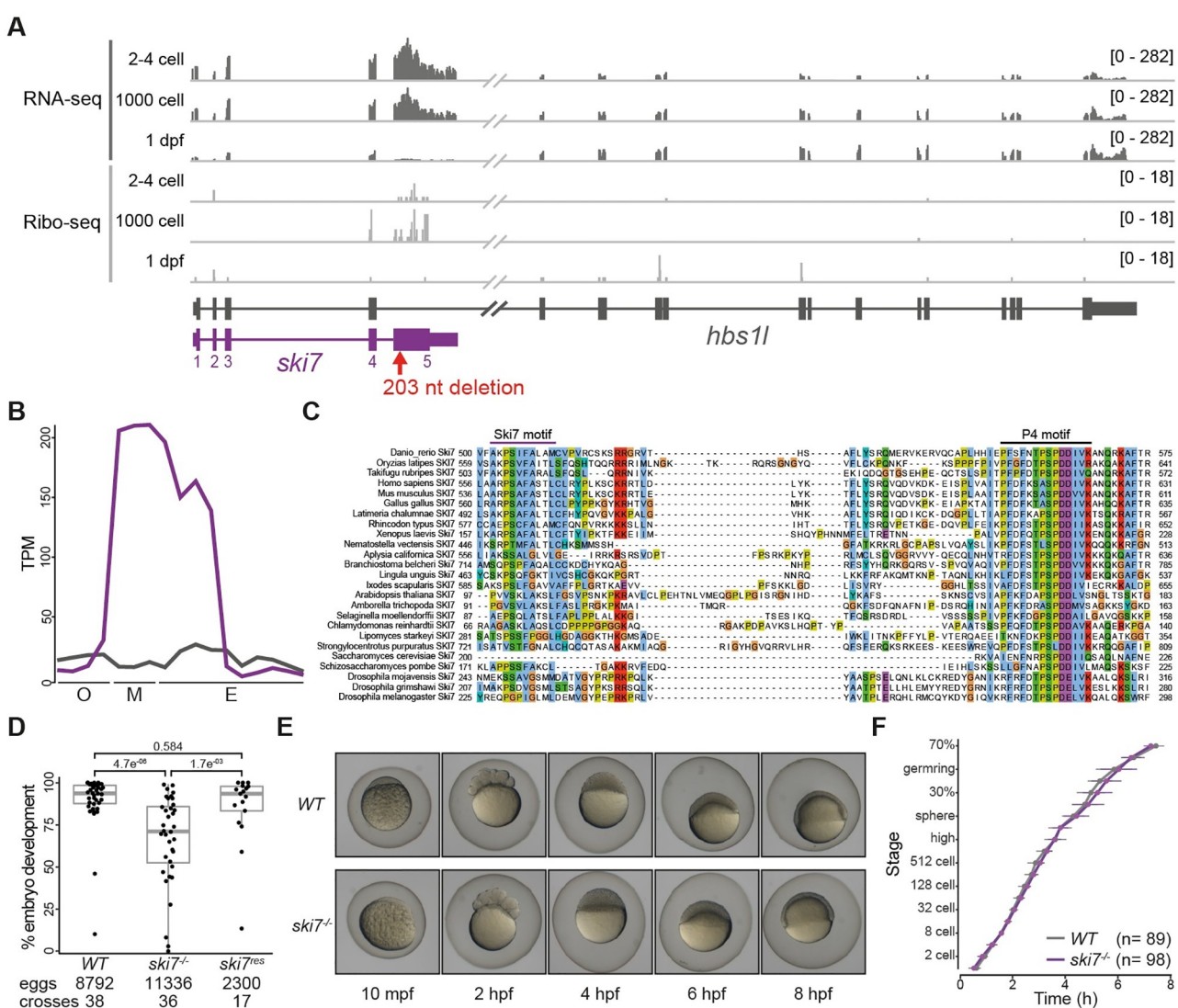

**Fig 1. Ski7 mutant fish produce reduced numbers of developing offspring.** (A) Expression profile of the *hbs1l* (gray)/*ski7* (purple) locus during embryogenesis. *Ski7* is highly expressed in the early embryo. RNA-seq is shown in dark gray and Ribo-seq in light gray; numbers in brackets indicate expression levels; the red arrow indicates the site of the *ski7* mutation. (B) *Ski7* mRNA peaks during the oocyte-to-embryo transition. Transcript-specific quantification of *ski7* (purple) and *hbs1l* (gray) RNA from polyA⁺ RNA-seq during oogenesis (O: oocyte stages I-IV), in mature eggs (M: unactivated, activated, fertilized) and during embryogenesis (E: 2–4 cell, 256 cell, 1000 cell, oblong, dome, shield, bud, 1 dpf, 2 dpf, 5 dpf [34]); TPM, transcripts per million. (C) Ski7 protein is highly conserved. Protein sequence alignment of the conserved C-terminus of Ski7 across different organisms. The two most conserved motifs, the Ski7-like motif and the Patch (P4)-like motif, are highlighted. (D) *Ski7⁻/⁻* fish produce fewer embryos that develop. Number of embryos that progress beyond the one-cell stage of wild-type (*WT*), *ski7⁻/⁻* and *ski7ʳᵉˢ* (*ski7⁻/⁻*; *tg[actb2:ski7-GFP]*) fish. Transgenic *ski7-GFP* rescues the decreased number of viable progeny observed in *ski7⁻/⁻* fish. P-adjusted values from Kruskal-Wallis with Dunn's multiple comparison test. (E) *Ski7⁻/⁻* embryos develop normally. Representative images of *WT* and *ski7⁻/⁻* embryos from 10 minutes post-fertilisation (mpf) to 8 hours post-fertilisation (hpf). (F) Quantification of embryo development in *WT* and *ski7⁻/⁻* embryos. *Ski7⁻/⁻* embryos that undergo cell cleavage develop at normal rate.

lack of an overt developmental phenotype suggest that Ski7's main function is during oogenesis and is important for organismal fitness during the oocyte-to-embryo transition.

## Zebrafish Ski7 interacts with the RNA exosome

Ski7 is known to interact with the cytoplasmic RNA exosome in yeast [16,27], plants [32], and human cells [24,25] via multiple interactions with the cap protein Csl4 by Ski7's highly

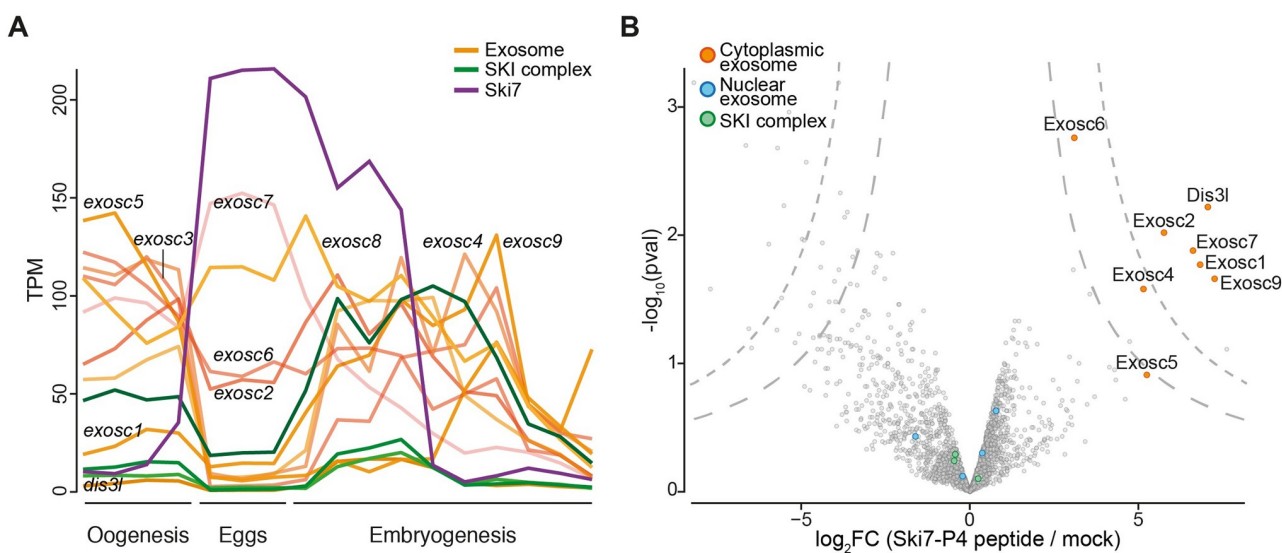

**Fig 2. Zebrafish Ski7 interacts with the cytoplasmic exosome.** (A) mRNA expression levels of *ski7* (purple), subunits of the cytoplasmic RNA exosome (orange) and the SKI complex (green). PolyA$^+$ RNA-seq during oogenesis (oocyte stages I-IV), mature eggs (unactivated, activated, fertilized) and embryogenesis (2–4 cell, 256 cell, 1000 cell, oblong, dome, shield, bud, 1 dpf, 2 dpf, 5 dpf [34]); TPM, transcripts per million. (B) Ski7 associates with the cytoplasmic exosome in early embryos. Volcano plot of Ski7-P4-like peptide-interacting proteins from wild-type zebrafish embryo lysates (128-512-cell stage) identified by mass spectrometry. Subunits of the cytoplasmic exosome (orange) are enriched, while nuclear exosome auxiliary factors (blue) and the SKI complex (green) are not enriched.

conserved P4-like motif. Transcriptomic analyses revealed that components of the zebrafish exosome are relatively lowly expressed during the oocyte-to-embryo transition (Fig 2A), leaving it unclear whether zebrafish Ski7 can interact with the exosome during this time frame. To determine whether the interaction of Ski7 with the exosome via the P4-like motif was conserved in zebrafish embryos, we conducted pull-down experiments using an *in vitro* synthesized, biotinylated zebrafish Ski7-P4-like peptide. Incubation of the peptide with early zebrafish embryo lysate followed by shot-gun mass spectrometry identified 7 out of 10 core components of the cytoplasmic exosome as top enriched Ski7-P4-like-interacting proteins (Fig 2B and S5 Table). These results suggest that zebrafish Ski7 is able to interact with the exosome in early embryos via its P4-like motif.

## Ski7 regulates transcript expression during the oocyte-to-embryo transition

As a first step towards characterizing Ski7's regulatory role during the oocyte-to-embryo transition, we compared the transcriptomes of wild type and *ski7*$^{-/-}$ mutants using RNA sequencing. Given that zebrafish Ski7 can interact with the cytoplasmic exosome (Fig 2B), mRNAs degraded in a Ski7-dependent manner were expected to be stabilized in *ski7*$^{-/-}$ mutants. To identify putative Ski7-dependent mRNA targets, polyA$^+$ RNA was isolated at eleven time points spanning the oocyte-to-embryo transition (period 1: oogenesis (O1 = I, O2 = II, O3 = III, O4 = IV & V [37,38]), period 2: mature eggs (UNA = unactivated, ACT = activated, FER = fertilized), and period 3: embryogenesis (E1 = 2-cell, E2 = 64-cell, E3 = 1000-cell, E4 = sphere) (Fig 3A and S6 Table). Principal component analysis (PCA) of transcript expression levels of wild type and *ski7*$^{-/-}$ mutants across this time series revealed that samples clustered primarily by developmental period (oogenesis vs. embryogenesis (PC1)) and time (early vs. late (PC2)) and not by genotype (Fig 3B). Due to the known large-scale gene expression

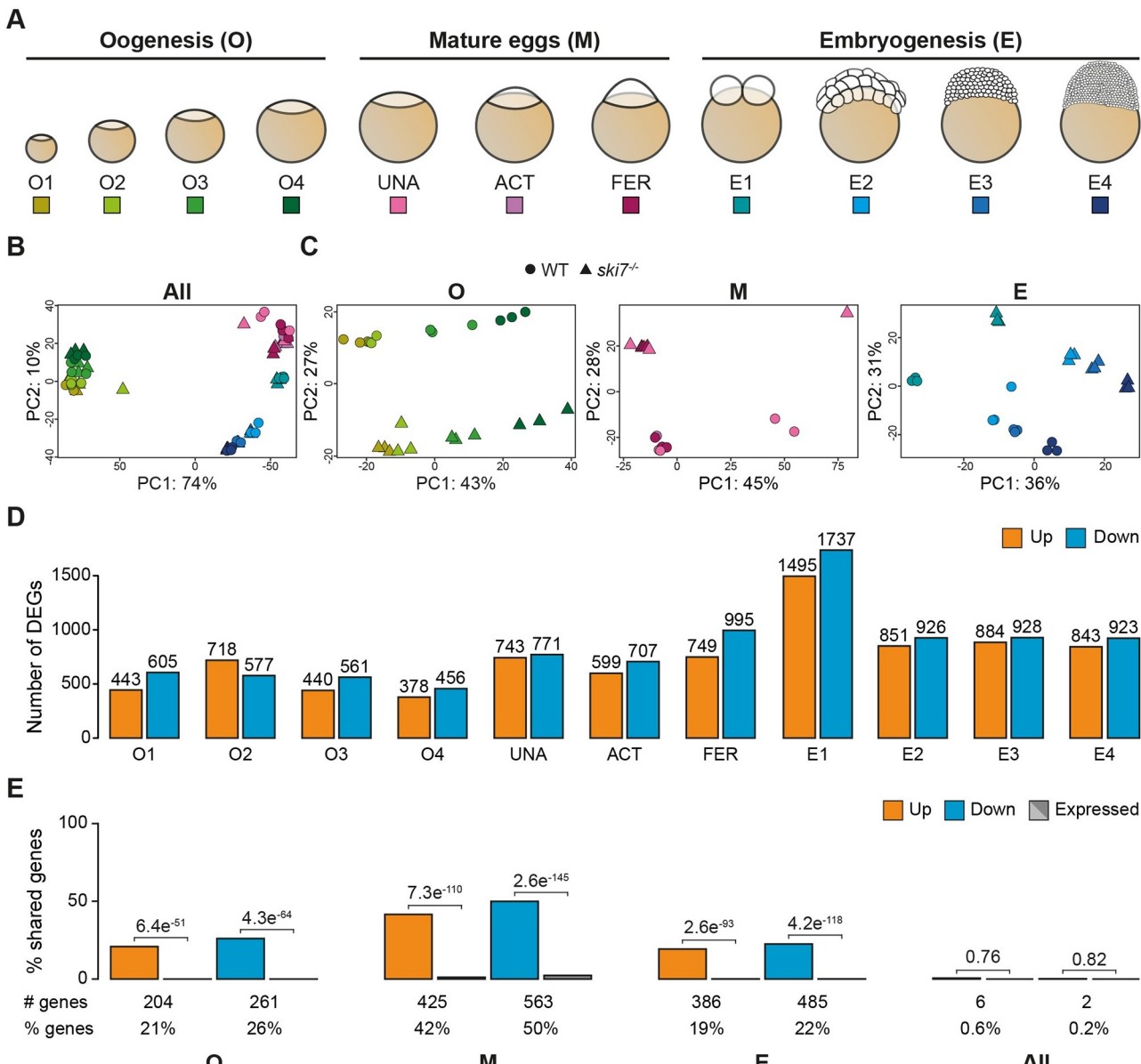

**Fig 3. Ski7 modulates the transcriptome during the oocyte-to-embryo transition.** (A) Schematic representation of the stages used for RNA-seq during three consecutive developmental periods: oogenesis, mature eggs and embryogenesis. (B, C) Principal Component Analyses (PCA) of all time points (B) or individual periods (C) used for the polyA+ RNA-seq comparison of *WT* (circle) and *ski7*<sup>-/-</sup> (triangle). (B) Samples show clustering by developmental time. (C) Samples within each period (oogenesis, mature eggs, and embryogenesis) are separated by time and genotype. The colour code corresponds to the different stages from Fig 3A. (D) Differentially expressed genes (DEGs) identified per stage based on polyA+ RNA-Seq data (orange: up-regulated in *ski7*<sup>-/-</sup> mutants; blue: down-regulated in *ski7*<sup>-/-</sup> mutants). (E) Percentage of shared DEGs (orange: up-regulated; blue: down-regulated) and expression-matched unchanged genes in wild type (gray) per period (oogenesis, eggs, embryogenesis) and during the full time-course (All). P-values from Pearson's chi-squared test with Yates continuity correction from comparison of the number of up- or down-regulated genes versus expression-matched genes.

changes between oogenesis and embryogenesis, we suspected that subtler differences between wild-type and *ski7*<sup>-/-</sup> mutant transcriptomes might be masked. To address this, we performed PCA for each of the three periods separately. This analysis revealed a clear separation by genotype for each period (Fig 3C), which suggests that *ski7*<sup>-/-</sup> mutants indeed differ in their transcriptomes from wild types during the oocyte-to-embryo transition.

Analysis of differentially expressed genes (DEGs) between wild-type and *ski7*$^{-/-}$ mutants identified 1942, 2151, and 4155 DEGs (FDR $\leq$ 0.05) during oogenesis, in eggs, and during embryogenesis, respectively (Fig 3D and S7–S11 Tables). Similarly, analysis of rRNA depleted (rRNA-) samples during oogenesis and embryogenesis identified 2791 and 2482 DEGs (FDR $\leq$ 0.05), respectively (S4A Fig and S7–S11 Tables). Around one third of DEGs was identified by both mRNA enrichment methods (S4B Fig), and gene expression differences between wild type and *ski7*$^{-/-}$ mutants showed overall high correlation (S5 Fig). Due to the consistent results between both mRNA enrichment methods, but the more comprehensive data set obtained from polyA$^+$ RNA spanning also the mature egg stages, we decided to focus our downstream analyses on the DEGs identified by polyA$^+$ RNA-Seq.

Comparison of the overlap of identified DEGs revealed a significant overlap of DEGs between individual time points within each period (19%-50%) compared to randomly selected, expression-matched unchanged genes (Figs 3E and S6A). This indicates that there is a core set of genes that are targeted by Ski7 during each period. However, DEGs shared amongst all three periods were rare, making DEGs largely period-specific (0.6% and 0.2% overlap of up- or down-regulated genes, respectively) (Figs 3E and S6B).

Overall, our results identified a higher than expected overlap of DEGs within periods, but little overall between periods. One possible reason for the sparsity of shared DEGs in *ski7*$^{-/-}$ mutants across the oocyte-to-embryo transition is that DEGs might be specifically enriched for genes that are exclusively expressed at a specific time period in wild types. Alternatively, DEGs may be expressed during more than one time period in wild types, but only differentially regulated during a defined time in *ski7*$^{-/-}$ mutants. To distinguish between these two possibilities, we assessed the wild-type expression dynamics of each DEG over time. We found that between 68% and 99% of Ski7 targets mis-regulated during only one period were expressed during more than one period (S7A and S7B Fig). In addition, when we compared DEGs from a specific period against unchanged genes from the other two periods, we observed that between 8% and 39% of DEGs were considered as unchanged (FDR > 0.05; 0.8 < FC < 1.2) (S8A and S8B Fig). This suggests that a subset of DEGs arise due to time-specific regulation by Ski7.

## Ski7 targets lowly expressed mRNAs for 3'-to-5' degradation

Analysis of the expression levels of DEGs in wild types revealed that genes upregulated in *ski7*$^{-/-}$ mutants were more lowly expressed in wild types than downregulated and unchanged genes (Fig 4A). This characteristic was consistently observed at each time point during the oocyte-to-embryo transition, which indicates that Ski7 normally targets transcripts with low steady-state levels for RNA degradation. For still unclear reasons we also found that in the mature egg down-regulated genes were more highly expressed in wild types (Fig 4A).

To address whether Ski7 targets differ in their read distributions along gene bodies in wild type versus *ski7*$^{-/-}$ mutants, we analyzed metagene profiles of up- and downregulated genes as well as unchanged genes for each period. We focused our analyses on the group of shared DEGs within each period (hereafter referred to as 'shared period DEGs') as they contain the most confidently identified Ski7-regulated genes per time-window. As expected, metagene profiles from unchanged genes showed no differences between wild types and *ski7*$^{-/-}$ mutants (Figs 4B and S9, and S12 Table). Additionally, consistent with genes being up- or downregulated in *ski7*$^{-/-}$ mutants, *ski7*$^{-/-}$ profiles displayed overall higher or lower read coverage compared to wild-type profiles, respectively (Figs 4B and S9, and S12 Table). Although this effect was observed across the full transcript, comparison of read densities across individual transcript regions (5' UTR, coding sequence (CDS), 3' UTR) revealed a relative enrichment of

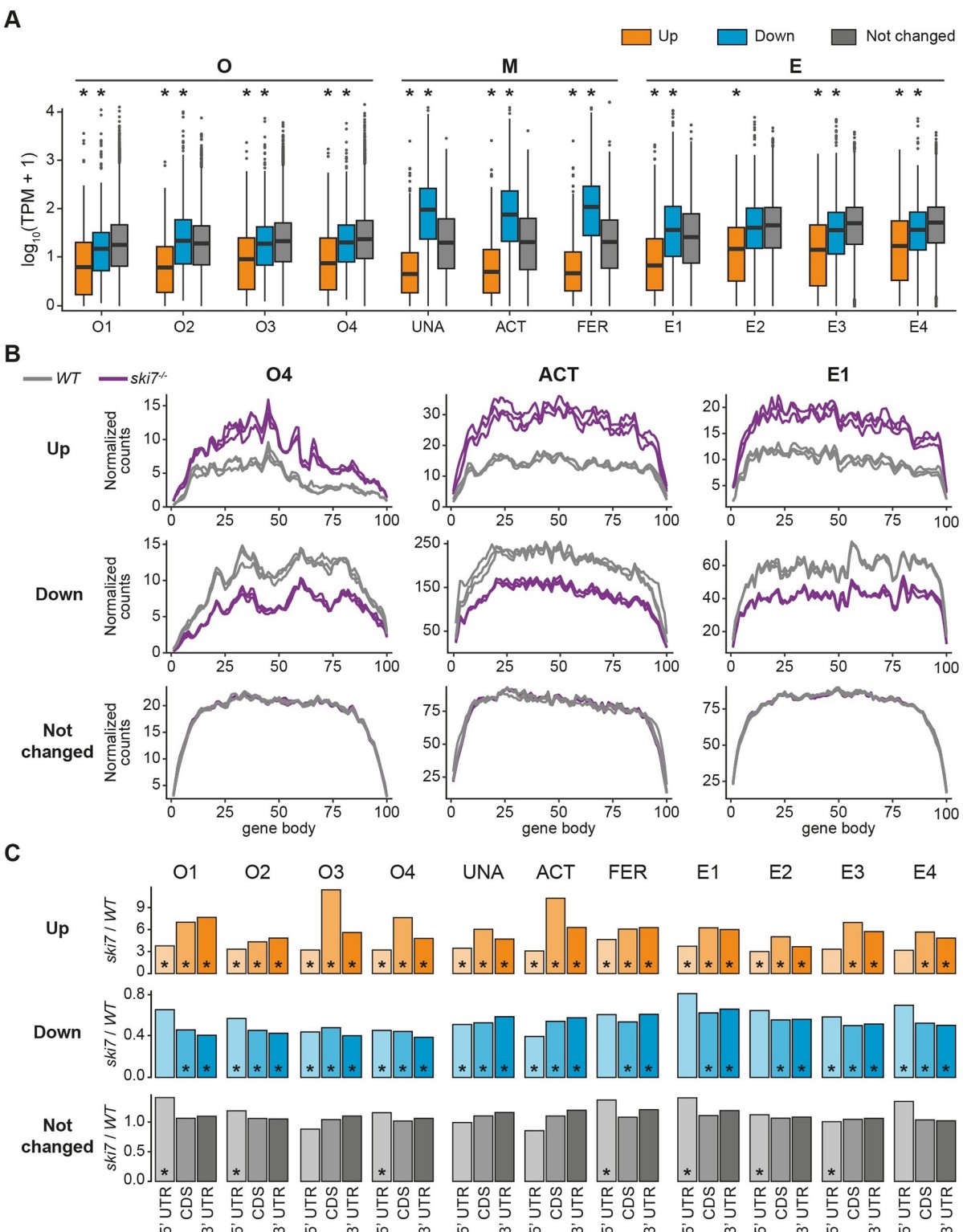

**Fig 4. Genes overexpressed in the *ski7* mutants are lowly expressed.** (A) Genes up-regulated in the absence of Ski7 show low expression levels in wild type. Expression levels of DEGs in wild type for every stage of the time course, as measured by transcripts per million (TPM). * indicates p-value < 0.01 from Wilcoxon test in comparison to unchanged genes. (B) Metagene profiles of up-regulated (top), down-regulated (middle), and unchanged genes (bottom) of wild type and *ski7^-/-* mutants at a representative stage for each period during the oocyte-to-embryo transition (oogenesis = O4, mature eggs = activated eggs, embryogenesis = E1). *WT*, gray; *ski7^-/-*, purple. (C) Up-regulated genes have

higher read density towards their 3' end. Ratio of read density of *ski7*[-/-]/*WT* per gene body region (5' UTR, CDS, 3' UTR) for all stages covered in this study. * p-value < 0.05; Wilcoxon test of read density of *ski7*[-/-] vs. read density of *WT* per region.

reads in the CDS and 3' UTR compared to the 5' UTR for up-regulated genes (Fig 4C). This bias towards the 3' end of up-regulated transcripts supports the idea that zebrafish Ski7 acts similarly to yeast Ski7 in contributing to 3'-to-5' mRNA degradation.

## Ski7 targets transcripts with diverse features

It has been previously shown that inherent RNA properties, such as transcript and polyA-tail length, codon usage and RNA structure influence RNA stability [39–42]. To investigate whether Ski7 targets share specific transcript features that make them prone for Ski7-dependent regulation, we assessed whether shared period DEGs differ in their intrinsic transcript properties compared to unchanged genes. We observed minor yet significant differences in gene length between the two groups. Genes down-regulated during oogenesis were overall shorter than unchanged genes, while genes up-regulated during the same period were longer (Kolmogorov-Smirnov p = $1.7e^{-07}$ & 0.003, for down- and up-regulated genes respectively) (S10A Fig and S13A Table). This difference could be attributed largely to the CDSes and 3' UTRs for down-regulated genes (S10B Fig and S13C and S13D Table) and to the longer 5' and 3' UTRs for up-regulated genes (S10B Fig and S13B and S13D Table). Additionally, we also observed significantly shorter 3' UTRs in up-regulated genes during embryogenesis (S10B Fig and S13D Table). In addition, to determine possible differences in the usage of synonymous codons, we calculated the codon adaptation index (CAI) [43] of the coding sequences of DEGs and unchanged genes. Genes up-regulated during oogenesis showed a lower CAI than down-regulated or unchanged genes (Kolmogorov-Smirnov, p = 0.004) (S11 Fig and S14 Table). This indicates an enrichment of rare codons in genes degraded in a Ski7-dependent manner in the female germline.

## Absence of Ski7 confers increased resistance to reductive stress

To determine whether Ski7-dependent mis-regulation of transcripts might result in an imbalance at the protein level, we performed tandem mass tag mass spectrometry (TMT-MS) with wild-type and *ski7*[-/-] embryos at four hours post-fertilization. While the overall amount of proteins did not differ in wild-type and *ski7*[-/-] embryos, 76 proteins were found to differ significantly between wild-type and *ski7*[-/-] mutant embryos (33 up-regulated, 43 down-regulated), of which 31 (15 up-regulated, 16 down-regulated) had also been detected as DEGs by RNA-seq (Figs 5A and S12A, and S7–S11 and S15 Tables).

We noticed that several proteins upregulated in the absence of Ski7 are implicated in the regulation of stress response and redox processes (Figs 5A and S12B). To determine whether mis-regulated genes belonged to common pathways or shared biological functions, we performed GO enrichment analyses of DEGs. Enriched terms in either up- or down-regulated shared period DEGs included processes associated with redox-related processes, cellular respiration, and cellular response to stress across all three periods (S13 and S14 Figs).

Because GO term enrichment analysis and our proteomic analysis indicated a possible link between Ski7 and redox regulation, we assessed whether *ski7*[-/-] embryos differed in their response to redox stress. Treatment of wild-type and *ski7*[-/-] embryos with the reducing agent DTT revealed that *ski7*[-/-] embryos were more resistant to reductive stress than wild-type embryos (Fig 5B and 5C and S16 Table). This effect persisted throughout the time course

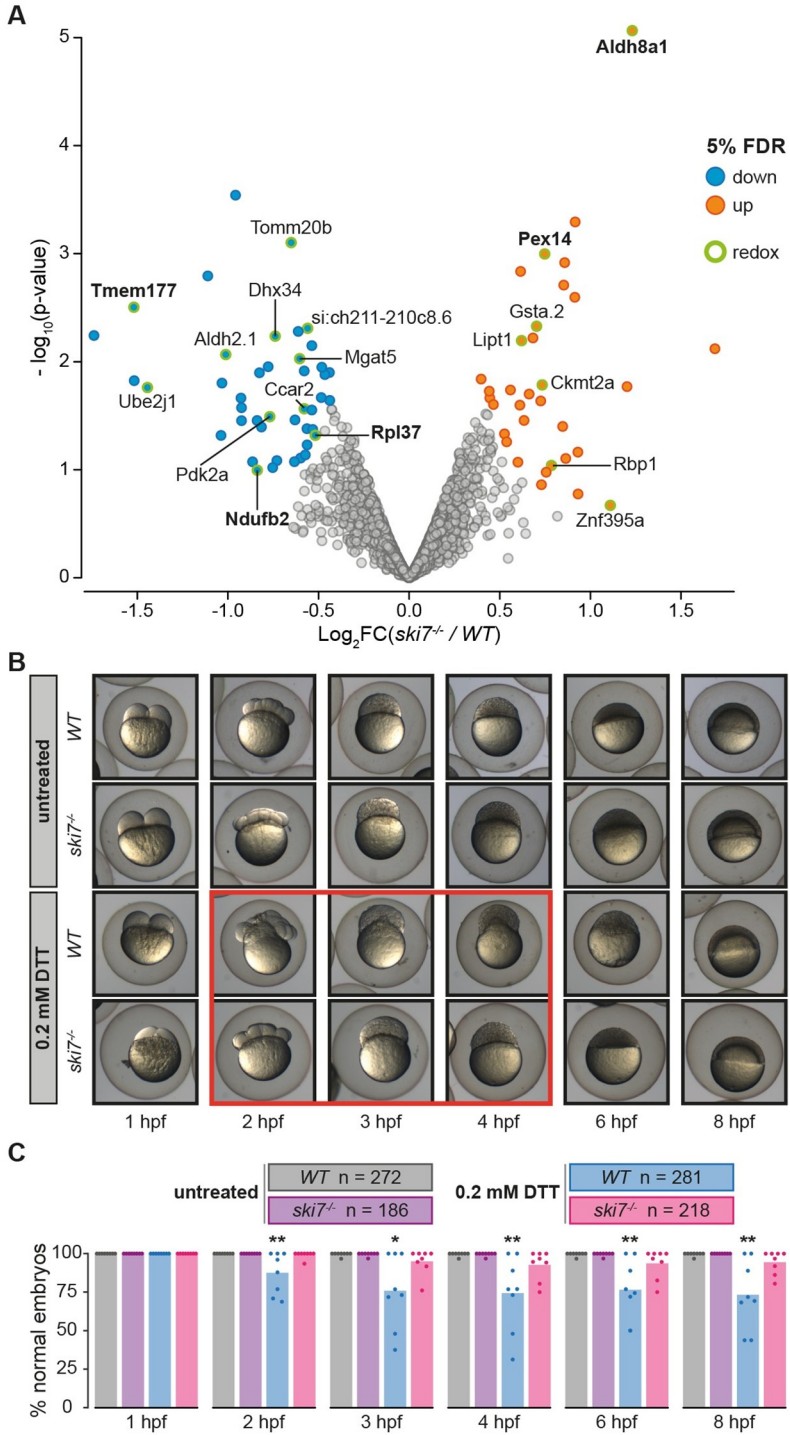

**Fig 5. Absence of Ski7 confers increased resistance to reductive stress.** (A) Volcano plot of proteins identified by tandem mass tag mass spectrometry (TMT-MS) from wild-type and *ski7$^{-/-}$* mutant embryos at 4 hours post-fertilization. Significantly up- and down-regulated proteins are coloured in orange and blue, respectively. Proteins associated with oxidation-reduction processes are highlighted (green circle) and their name is displayed. Proteins, whose RNA was also identified as differentially expressed based on RNA-seq (DEGs), are highlighted in bold. (B) *Ski7$^{-/-}$* mutant embryos show increased resistance to DTT-treatment. Representative images of the development of wild-type and *ski7$^{-/-}$* embryos incubated in the absence (untreated) or presence of 0.2 mM DTT. The red box highlights the time-frame during which wild-type embryos show the most obvious aberrant morphogenesis. (C) Quantification of developmental abnormalities upon treatment with 0.2 mM DTT during embryogenesis (hpf, hours post-

fertilization). Each dot represents the quantification of an independent experiment (* = p-adjusted < 0.05; ** = p-adjusted < 0.01 from Kruskal-Wallis with Dunn's multiple comparison test compared to *WT* untreated on each time point).

(Fig 5B and 5C and S16 Table). Overall, these results suggest that embryos lacking Ski7 can cope better with reductive stress, potentially by having elevated levels of redox-regulating factors during the oocyte-to-embryo transition.

## Discussion

In this study, we provide the first functional characterization of vertebrate Ski7 and the consequences of its absence. Zebrafish Ski7, similar to its human and plant homologs [24,25,32], is encoded as an alternative splice isoform of *hbs1l*. We found that zebrafish *ski7* but not *hbs1l* mRNA is highly expressed in mature eggs and early embryos. Our phenotypic characterization of zebrafish *ski7*[-/-] mutants revealed that *ski7*[-/-] mutants have compromised egg quality and produce fewer developing embryos (Figs 1D and S3A and S3B, and S2 and S3 Tables), which in a natural environment could result in decreased fitness and ultimately lead to impaired propagation and maintenance of the species. The lack of progression beyond the one-cell stage in *ski7*[-/-] mutants, together with our finding that, once they passed the first cell division, *ski7*[-/-] mutant embryos have no further apparent defects in development, suggests that Ski7 mainly acts during oogenesis and/or in the mature egg.

What could be the molecular basis for Ski7 affecting oogenesis, fertility, or failure to undergo the first cell cleavage? A possible explanation could be that Ski7 directly affects transcripts involved in oocyte maturation or fertilization. Although we have evidence that certain transcripts regulated by Ski7 correspond to fertilization associated processes based on GO terms (egg coat) (S14 Fig), they did not comprise the majority of affected genes. Moreover, the transcript levels of *bouncer*, the only known fertility factor in fish eggs [44], was not decreased in *ski7*[-/-] mutants. In addition, visual inspection of the micropyle, the funnel-like structure through which the sperm access the egg membrane, revealed no defect in the formation of this structure in *ski7*[-/-] mutant eggs (100% of good-quality *ski7*[-/-] mutant eggs had a visible micropyle). Given that a higher proportion of *ski7*[-/-] mutant eggs than of wild-type sibling eggs tended to be of poor quality, we think that the presumable subfertility/early embryo development defect is more likely to be secondary to a mis-regulation of the transcriptome during oogenesis. Furthermore, DEGs in *ski7*[-/-] mutant oocytes and eggs partially overlapped with genes shown to be mis-regulated in poor-quality zebrafish eggs [45,46], further supporting our conclusion that mis-regulation of the transcriptome in *ski7*[-/-] mutants might contribute to the poor quality of the eggs.

In analogy to the situation in yeast [14,16] and human cells [24,25], we have two lines of evidence that zebrafish Ski7 acts in conjunction with the cytoplasmic RNA exosome in promoting 3'-to-5' RNA decay: First, we find that zebrafish Ski7 can interact with the RNA exosome (Fig 2B). And second, up-regulated genes show an accumulation of reads towards their 3' ends (Fig 4C). Our conclusion that the Ski7/RNA exosome complex acts during zebrafish egg development is consistent with the recently reported role for the exosome core component EXOSC10 in the maturation of eggs in mouse [47] and the Ski complex in *Drosophila* [48]. While the phenotypic consequences of loss of Ski7 in mice remain to be determined, our analysis of publicly available RNA-seq data [49,50] revealed that mouse *Ski7* mRNA is also highly expressed in mature eggs and decreases during embryogenesis while *Hbs1l* remains stable (S1C Fig). Taken together, Ski7's function in fine-tuning transcript levels during egg maturation by facilitating 3'-to-5' RNA degradation might be conserved in vertebrates.

Our data reveal that Ski7 preferentially targets transcripts for degradation whose steady-state levels are low (Fig 4A). This suggests a requirement for Ski7 in maintaining low transcript abundances for certain genes and/or reducing transcriptional noise, possibly in conjunction with other mechanisms including the preferential use of rare codons. While we expected transcripts to become predominantly up-regulated in *ski7* mutants, many genes were also found to be down-regulated (Fig 3D and S7–S11 Tables). Down-regulated genes could either be due to secondary effects caused by the lack of down-regulation of primary Ski7/RNA exosome target genes, or due to so-far unknown activities of Ski7. Interestingly, down-regulated genes in mature eggs were biased towards being more highly expressed in wild type (Fig 4A). Future work will be needed to address whether these transcripts might be subjected to ribosome stalling in the absence of Ski7, and thus be targeted for degradation by alternative pathways, as we also observed accumulation of transcripts related to translation in the mature egg and embryo (S13 and S14 Figs). While we have currently no evidence for alternative functions for Ski7, an alternative possibility is that Ski7 might interact with other factors in addition to the RNA exosome, particularly in light of the comparatively low mRNA expression levels of the core exosome components during the oocyte-to-embryo transition (Fig 2A). However, our finding that a Ski7-P4-like peptide fragment can stably bind to the zebrafish exosome complex in early embryo lysates demonstrates that zebrafish Ski7 can stably interact with the RNA exosome (Fig 2B and S5 Table). This experiment also shows that the zebrafish Ski7-P4-like motif (PFSFNTPSPDDIVK) is sufficient to pull down the exosome in zebrafish embryos. This is in contrast to data from human cells in which the Ski7-like motif but not the P4-like motif was shown to be sufficient for mediating this interaction [25]. Thus, our data demonstrates a direct interaction between Ski7 and the RNA exosome.

Another question that will warrant further investigations is the underlying reason for the relatively small number of shared DEGs among different developmental periods. Along these lines, several recent studies have challenged the view that the general RNA decay machinery is unspecific. For example, Dcp2 (5' decapping) and CCR4NOT (3' deadenylation) target specific subsets of mRNAs depending on inherent transcript features during zebrafish embryo development [51–56]. However, it remains largely unclear to what extent general cytoplasmic RNA degradation machineries or specific components thereof contribute to the large-scale transcriptome remodeling during development. Our analyses indicate that the lack of shared DEGs between periods is not due to time-restricted expression of the target mRNAs in the wild-type situation but rather time-specific regulation by Ski7. How Ski7's time-specificity is achieved is still unclear. Possible mechanisms include temporal stabilization of transcripts by RNA-binding proteins or sequestration from the Ski7/RNA exosome complex, as well as yet-to-be-identified temporal Ski7-specific regulators.

Our observation that *ski7*[-/-] mutant embryos are less sensitive to reductive stress was unexpected since previous studies in yeast have indicated the opposite effect: In yeast, Ski7-mediated non-stop decay (NSD) was shown to be required for oxidant tolerance in combination with the Dom34-Hbs1 complex [57]. In mammals, it has been shown that redox regulation, e.g. by regulating the levels of NADPH/NADH, is particularly important during oogenesis and in the egg [58–61]. Such de-regulation of NADH/cellular respiration in *ski7*[-/-] eggs could also contribute to the observed reduced number of developing embryos in *ski7*[-/-] mutants, in addition to the mis-regulation of reproductive genes themselves (S14 Fig). Consistent with this idea, enriched GO terms for downregulated genes included oxidoreductase activity, cellular respiration, and cellular response to stress (S13 and S14 Figs). These pathways have been recently shown to be important for the generation of high quality oocytes in mammals [59–61].

There are several possible explanations for the increased tolerance of ski7[-/-] mutants to the reductive agent DTT during embryogenesis. In the simplest scenario, Ski7 might negatively regulate redox stress tolerance in zebrafish embryos by fine-tuning the levels of embryonic redox regulators. However, the contrasting observations, namely that ski7[-/-] mutant eggs produce fewer embryos that develop beyond the one-cell stage, but that mutant embryos are more resistant to reductive stress, are difficult to explain by a static scenario. We therefore propose an alternative possibility, namely that Ski7 might contribute to the dynamic regulation of redox balance during the oocyte-to-embryo transition. In support of this idea, we found that cellular respiration and mitochondrial processes are mis-regulated throughout the oocyte-to-embryo transition in ski7[-/-] mutants, which could result in a time-dependent imbalance in redox regulation. Evidence for increased oxidative levels after fertilization come for example from *Xenopus*, in which fertilization enhances the production of mitochondrial ROS [62], *C. elegans* [63] and zebrafish embryos [64]. Additionally, in support of the idea of dynamic redox regulation by Ski7, it was reported that *Drosophila* oocytes lacking an oocyte-specific thioredoxin had increased levels of $H_2O_2$ and were more resistant to DTT than their wild-type siblings [65]. Apart from the idea that Ski7 could regulate the dynamic redox balance throughout the oocyte-to-embryo transition, another, non-contradicting possibility is that the increased resistance to reductive agents during embryogenesis might be a direct or indirect consequence of compromised egg maturation in the absence of Ski7. As such, counter-measures of the egg combatting the impaired regulation of RNA levels during oogenesis might make the egg better prepared for later encounters of redox stresses. As such, up-regulation of redox-regulators and/or chaperones during oogenesis could enable the embryo to deal better with reductive agents. This is in line with previous findings that mammalian oocytes treated with $H_2O_2$ produced more blastocysts under *in vitro* conditions [66–68]. While we currently don't know the extent or direct impact of Ski7-mediated regulation on the redox balance during the oocyte-to-embryo transition, one intriguing speculation is that Ski7 production itself might be redox-controlled. Splicing factors have been shown to be differentially regulated by redox stress [69,70], opening the possibility that the alternative splicing reaction responsible for generating the Ski7-specific splicing event might at least in part be triggered by the redox-conditions encountered during oogenesis.

Overall, our data supports the idea that Ski7 is important for oogenesis, and for producing eggs that can be fertilized and successfully undergo the first cell cleavage, by fine-tuning the 3'-to-5' RNA degradation machinery, namely the RNA exosome, to aid in the timely and controlled transition from the oocyte to the embryo.

## Materials and methods

### Ethics stament

All fish experiments were conducted according to Austrian and European guidelines for animal research and approved by the Amt der Wiener Landesregierung, Magistratsabteilung 58—Wasserrecht (animal protocol GZ: 342445/2016/12).

### Fish husbandry

Zebrafish (*Danio rerio*) were raised according to standard protocols (28˚C water temperature; 14/10 hour light/dark cycle). TLAB fish, generated by crossing zebrafish AB and the natural variant TL (Tupfel Longfin) stocks, served as wild-type zebrafish for all experiments. The ski7[-/-] mutant line and the transgenic Ski7-rescue line were generated in the TLAB background and are described below.

## Conservation analysis

The last 228 nucleotides (76 amino acids) of the zebrafish Ski7-specific exon (exon 5) were used to perform the conservation analyses. This region included the predicted Ski7-like and P4-like motifs. The zebrafish Ski7 sequence was taken as a query for iterative PSI-BLAST searches against the NCBI non-redundant protein data base. In the first search round, not only vertebrate orthologs but also sequences from the coral *Stylophora pistillata* and the brachiopod *Lingula unguis* were detected (E-values < 0.001). Further iterations extended to the plant (e.g. *Ananas comosus*, round 2, E-value 2.17e-05) and fungi kingdom (e.g. *Botryotinia calthae*, round 3, E-value 3.73e-09). Significant hits covered both motifs. However, sequences with a weak Ski7-like motif, such as *Drosophila melanogaster* or *Saccharomyces cerevisiae* could not be identified with zebrafish Ski7, but were found with iterative PSI-BLAST searches of related insect or fungi orthologs. Sequences that represented a wide taxonomic range were selected, aligned using mafft (version 7.427 [71]) and visualized in Jalview [72].

## Generation of *ski7*<sup>-/-</sup> Mutant Fish

Zebrafish *ski7*<sup>-/-</sup> mutants were generated by CRISPR/Cas9-mediated mutagenesis. CHOP-CHOP [73,74] was used to predict three guide RNAs targeting the *ski7*-specific exon 5:

| | |
|---|---|
| **gRNA 1** : | GCACTGCTAACAGATCACTGAGG |
| **gRNA 2** : | GCCAAACTTCCAAAACCAGGGGG |
| **gRNA 3** : | AGACCAGTGGAGGGAACTAGAGG |

T7-promoter sequence- and 3' overhang-containing DNA oligos were synthesized (Sigma-Aldrich), annealed to a common tracer oligo (AAAAGCACCGACTCGGTGCCACTTTTT CAAGTTGATAACGGACTAGCCTTATTTTAACTTGCTATTTCTAGCTCTAAAAC), and *in vitro* transcribed by T7 polymerase (Ambion MEGAscript T7 transcription kit) according to a published protocol [75]. In-house produced Cas9 protein and pooled gRNAs were co-injected in the cell of one-cell TLAB zebrafish embryos. Potential founder fish were outcrossed to TLAB wild-type fish. Progeny were screened for carrying a mutation in *ski7* by PCR based on a size difference of the PCR product compared to the wild-type PCR product, using the primers ski7_F: GACTGTATCCAGTGCACATTCA and ski7_R: TTAAAAGAGCCAA GAG GACTGG. Embryos with potential mutations were raised. Adult fish were genotyped by standard fin-clipping procedures. Sanger sequencing of the PCR fragment identified a deletion of 214 nucleotides and an insertion of 11 nucleotides in *ski7's* terminal exon 5, resulting in a net deletion of 203 nucleotides that causes a frameshift mutation. Heterozygous fish were incrossed to generate homozygous *ski7*<sup>-/-</sup> mutant fish. The PCR products were detected by 2% gel electrophoresis (WT = 330 nucleotides, *ski7*<sup>-/-</sup> = 127 nucleotides).

## Generation of a transgenic *ski7*-GFP line

For the generation of the *ski7*-GFP transgenic line, the predicted *ski7* ORF was amplified from cDNA generated from 2–4 cell stage zebrafish embryos using the primers ski7_cDNA_F: GTGATTGCTAATCTTACTTTGAATTTGTTTACAGGGATCC GCGGCTGTGGAGTAA GACGCATG and ski7_cDNA_R: CTGGATCATCATCGTAC CATGGTTTGCTAGCGG CAGTCGAAGTGCTTTAAATAATTACACTG. The PCR fragment containing *ski7* was cloned into a vector containing Ampicillin resistance and *Xenopus* globin UTRs via SpeI/XhoI restriction sites, using the Gibson assembly method [76]. To generate C-terminally tagged *ski7*-GFP, the *ski7* ORF lacking the STOP codon was amplified from the previous vector with ski7_gfp_F: TAAACGCTCAACTTTGGCAGATCC and ski7_gfp_R:

GAACAGCTCCTCTCCTTTA GACACCATGGCTCCAGATCCGCTTCCAGAGTCTC GAGTAAAAGCTTTTCTCTGATTGG, and cloned via the Gibson assembly method [76] into a vector containing a Ser-Gly-linker followed by sfGFP, using SpeI/NcoI restriction sites. Subsequently, C-terminally *sfGFP-tagged ski7* was amplified by PCR and subcloned via BamHI/ EcoRV sites into a vector for Tol2-based transgenesis containing the *actb2* promoter and a *cmlc2-GFP* transgenesis marker (*Tol2*:*actb2-Ski7-SG-sfGFP*, *cmlc2GFP*).

This plasmid (~15 pg) was injected together with 35 pg Tol2 mRNA into the cell of 1-cell stage TLAB embryos. Putative founders were crossed to the *ski7*$^{-/-}$ line, and transgenic offspring were identified with the help of the cmlc2-GFP transgene (green hearts). To obtain rescued *ski7*$^{-/-}$ mutant fish (*ski7*$^{res}$: *ski7*$^{-/-}$; *tg[actb2:ski7-GFP]*), heterozygous *ski7*$^{+/-}$; *tg[actb2: ski7-GFP])* fish were incrossed to obtain homozygous *ski7*$^{-/-}$; *tg[actb2:ski7-GFP])* fish.

### Quantification of developing embryos

To quantify the proportion of developing embryos, individual male and female fish were set up together in breeding cages the night before mating. The next morning, fish were put together by removing the divider and allowing fish to mate for approximately 30 minutes. Eggs were collected from individual tanks and kept at 28˚C in blue water (E3 medium with 0.1% methylene blue). 'Poor quality' eggs (see below) were removed immediately from the dish. Only 'good quality' eggs (morphologically normal-looking, clear eggs that had successfully undergone egg activation as evident by normal chorion elevation) were considered for determining the fraction of eggs that developed beyond the one-cell stage. Embryo progression was determined between 5 and 6 hours post-fertilization.

### Quantification of egg quality

To quantify egg quality, individual male and female fish were set up together in breeding cages the night before mating. Wild-type and *ski7*$^{-/-}$ sibling females were crossed to wild-type male fish. Fish were put together by removing the divider and allowing fish to mate for approximately 30 minutes. Eggs were collected from individual breeding cages and moved to E3 medium with 0.1% methylene blue at 28˚C. Eggs were scored under a dissection microscope. Eggs were considered as "good" if they were normal-looking one-cell stage embryos (clear separation of yolk and cell; normal egg activation and chorion swelling; non-opaque appearance). Eggs were considered as "poor" if they were opaque, deformed, and no clear cell or chorion elevation was visible.

For the quantification of the presence of the micropyle structure, wild-type and *ski7*$^{-/-}$ sibling females were squeezed to obtain mature eggs. Eggs were kept in sorting medium at all times (Leibovitz's pH 9.0, 0.5% BSA), individually placed in an agarose mold and inspected for the presence of the micropyle under a dissection microscope.

### Phenotypic analyses and imaging

Quantification and staging of embryos were performed by live cell imaging on a Celldiscoverer 7 automated microscope (Zeiss). Wild-type and *ski7*$^{-/-}$ embryos were dechorionated with pronase (1 mg/ml). Dechorionated embryos were mounted in 0.5% low melt agarose within the first 30 minutes post-fertilization. Mounted embryos were kept in six-well plates in E3 medium with 0.1% methylene blue at 28˚C during imaging. Images were adquired every 5 minutes during the first 7 hours. Quantification and staging during the first hour post-fertilization was performed manually every five minutes under a stereo microscope at 28˚C.

For adult phenotyping, several adults (males and females) were manually inspected for morphological defects. Pictures were taken under a dissection microscope and a Blackfly S USB 3 camera (serial number 18255235) using the FlyCapture2 software.

## Ski7-P4-like peptide pull-down and MS analysis

The zebrafish Ski7-P4-like peptide (HHIEPFSFNTPSPDDIVKANQRK) was synthesized *in vitro* and N-terminally biotinylated (Biotin-Ahx-HHIEPFSFNTPSPDDIVKANQRK). The peptide contained the homologous region of yeast Ski7 that mediates the interaction with the Csl4 exosome protein (second half of helix 3 and first half of helix 4 in yeast) [24]. To prepare zebrafish embryo lysates, embryos were dechorinated with pronase (1 mg/ml). About 250 embryo caps per sample (each experiment was performed in triplicates) were manually dissected from 128-cell to 512-cell stage wild-type zebrafish embryos and immediately frozen in liquid nitrogen. Embryo caps were homogenized with a plastic pestle in 1 ml of cold binding buffer (150 mM KaOAc, 300 µl Hepes-NaOH pH 7.4, 5 mM $MgSO_4$, 0.1% NP-40, 5mM DTT, complete EDTA-free protease-inhibitors) and cleared by 1 minute centrifugation at 15000 rpm at 4˚C. The biotin pull-down was performed according to a protocol adapted from [77]. In brief, 30 µl of Streptavidin Dynabeads (MyOne, Invitrogen) per sample were washed with PBS and binding buffer. Beads were incubated with 2 µg of biotinylated Ski7-P4-like peptide in 500 µl of cold binding buffer (2 hours at 4˚C). Three samples of beads not incubated with biotinylated peptide served as negative controls for the pull-down. After washing of the beads (five times with 500 µl of binding buffer; total wash time 2 hours at 4˚C), beads were incubated with 1 ml embryo lysate for 2 hours at 4˚C on a rotating wheel. Beads were placed on a magnetic stand and washed six times with 1 ml of cold binding buffer (total wash time 2–4 hours at 4˚C). After washing, bound proteins were eluted by adding 30 µl of hot (95˚C) 2x SDS-PAGE sample buffer to the beads. For mass-spec analysis of the eluates, samples were run into a SDS-PAGE gel for a short time (without allowing separation of the proteins by size) and stained by Coomassie Blue. Gel elution of the Coomassie-stained band and tryptic digest followed standard procedures.

For LC–MS/MS, digested peptides were separated using a Dionex UltiMate 3000 HPLC RSLC nano system (Thermo Fisher Scientific) coupled to a Q Exactive mass spectrometer (Thermo Fisher Scientific), equipped with a Proxeon nanospray source (Thermo Fisher Scientific). Peptides were loaded onto a trap column (PepMap C18, 5 mm × 300 µm ID, 5 µm particles, 100 Å, Thermo Fisher Scientific) at a flow rate of 25 µL min$^{-1}$ using 0.1% TFA as mobile phase. After 10 min, the trap column was switched in line with the analytical column (PepMap C18, 500 mm × 75 µm ID, 2 µm, 100 Å, Thermo Fisher Scientific). A 105/165/225 min gradient from buffer A (water/formic acid, 99.9/0.1, v/v) to B (water/acetonitrile/formic acid, 19.92/80/0.08, v/v/v) was applied to elute the peptides at a flow rate of 230 nl min$^{-1}$. The Q Exactive HF mass spectrometer was operated in data-dependent mode, using a full scan (m/z range 380–1500, nominal resolution of 60,000, target value 1E6) followed by MS/MS scans of the 10 most abundant ions. MS/MS spectra were acquired using normalized collision energy of 27%, isolation width of 1.4 m/z, resolution of 30.000 and the target value was set to 1E5. Precursor ions selected for fragmentation (excluded charge state 1, 7, 8, >8) were put on a dynamic exclusion list for 20/40/60 s, depending on the gradient length. Additionally, the minimum AGC target was set to 5E3 and the intensity threshold was calculated to be 4.8E4. The peptide match feature was set to preferred and the exclude isotopes feature was enabled. For peptide identification, the RAW files were loaded into Proteome Discoverer (v2.1.0.81, Thermo Scientific). All MS/MS spectra were searched using MSAmanda v2.1.5.8715 [78] against a custom *Danio rerio* protein database (based on GRCz10 and CRC64 non-redundant Ensembl 86 release: 42,188

sequences, 22,758,666 residues). The following search parameters were used: fixed modification: beta-methylthiolation on cysteine; variable modifications: oxidation on methionine, phosphorylation on serine, threonine and tyrosine, deamidation on asparagine and glutamine, acetylation on lysine, methylation on lysine and arginine, di-methylation on lysine and arginine, tri-methylation on lysine, ubiquitinylation on lysine, glycosylation (HexNac) on asparagine, serine and threonine as well as glutamine-to-pyro-glutamic-acid-transformation on peptide-N-terminal glutamine. Monoisotopic masses were searched within unrestricted protein masses for tryptic enzymatic specificity. The peptide mass tolerance was set to ±5 ppm and the fragment mass tolerance to ±0.03 Da. The maximal number of missed cleavages was set to two. The result was filtered to 1% FDR on peptide level using Percolator algorithm integrated in Thermo Proteome Discoverer. The localization of the modification sites within the peptides was performed with the tool ptmRS, which is based on the tool phosphoRS [79]. Label-free quantification of peptides was performed using the in-house developed tool apQuant [80]. Proteins were quantified by summing up all peptides before performing subsequent iBAQ normalization [81]. Differentially expressed proteins were determined using limma [82]. P-values were adjusted for multiple testing as implemented in the limma R package [83].

## RNA isolation and sequencing

Total RNA was extracted using the standard TRIzol (Invitrogen) protocol. Samples from individual stages of three different periods (oogenesis, mature eggs, embryogenesis) were collected as follows: To obtain different stages of oogenesis, ovaries were dissected from three different females from each genotype. Wild-type and *ski7⁻/⁻* mutant females were "purged" in order to enrich for earlier stages of oogenesis, as described in [37]. Oocytes were manually sorted based on differences in size and opacity as previously described in [37] (O1 –smallest size, most translucent; O2 –still translucent but bigger than O1; O3 –bigger than O2, becoming opaque and with visible central germinal vesicle; and O4 –largest size but smaller than mature eggs, opaque) and kept in oocyte sorting medium (Leibovitz's medium, plus 0.5% BSA, pH 9.0 adjusted with 5 M NaOH [37]). To obtain different stages of mature eggs, unactivated, activated and fertilized eggs were collected from the same female fish (note: only eggs of good quality were collected). Three independent biological replicates were collected of each stage (three different females, between 35 to 60 eggs per condition). Briefly, single female fish were put together with male fish for standard mating. Eggs that were laid within the first minute after putting them together were collected at 10 minutes post-fertilization and homogenized in TRIzol (fertilized eggs). Fertilization of the eggs was quantified after 3 hours and only samples in which the rate of fertilization was higher than 60% were used. The females used for collecting fertilized eggs were immediately anesthetized using 0.1% Tricaine and subjected to squeezing to collect the remaining two stages, unactivated and activated eggs. Half of the squeezed eggs were immediately homogenized in TRIzol (unactivated eggs). The remaining half of the eggs was activated by adding fish water (E3 medium with 0.1% methylene blue) and incubated for 10 minutes before collection and homogenization in TRIzol (activated eggs). To obtain different stages during embryogenesis, between 20 to 30 embryos per time point were collected at the indicated stages and homogenized in TRIzol. Total RNA was isolated and assessed for quality and quantity based on analysis on the Fragment Analyzer. For PolyA+ RNA selection, 2 μg of total RNA was used per sample as input for the poly(A) RNA Selection Kit (LEXOGEN). For rRNA depleted samples, 1 μg of total RNA was used as input for the RiboCop rRNA Depletion Kit (LEXOGEN). Stranded cDNA libraries were generated using NEBNext Ultra Directional RNA Library Prep Kit for Illumina (New England Biolabs) and indexed with

NEBNext Multiplex Oligos for Illumina (Dual Index Primer Set I) (New England Biolabs). Library quality was checked on the Fragment Analyzer and sequenced on a Illumina Hiseq 2500 with the SR100 mode.

## RNA-seq data processing

RNA-seq reads were processed according to standard bioinformatic procedures. First, BAM files were converted to fastq files using samtools (v1.9) [84]. Barcoded libraries were demultiplexed using fastx-toolkit (v0.0.14) (http://hannonlab.cshl.edu/fastx_toolkit/) and customized perl scripts. Sequencing adapters were trimmed with bbduk command from the BBmap package (v38.26) [85] and only reads longer than 20 bases were kept for downstream analyses. Filtered reads were mapped to Ensembl 92 gene models (downloaded 2018.08.01) and the GRCz11 *Danio rerio* genome assembly, with Hisat2 v2.1.0 [86] (—rna-strandness R -k 12 –no-unal). Quantification at the gene level (transcript per million (TPM)) was perfomed using Kallisto (v0.43.0) [87].

## Differential gene expression analyses

HTSeq (v0.9.1) [88] was employed to count the uniquely mapped reads. Differential gene expression was performed on raw counts using EdgeR (v1.22.2) [89] with default parameters and size factors estimated from global normalization. The three different periods were analyzed separately, except for the generation of the combined PCA plot (Fig 3B). Only genes that had at least 10 cpm (counts per million) in at least three libraries per period were considered for the analyses. Genes were considered as differentially expressed if they had a FDR (false discovery rate) < 0.05. Unchanged genes were defined as FDR > 0.05 and fold change between 0.8 and 1.2. Principal component analyses were performed after a regularized log transformation using the 1000 most variable genes.

## Analyses of RNA features

For all analyses, only the longest isoform was used for genes with multiple isoforms. For metagene profiles, full length transcripts of at least 100 nucleotides were used. BAM files containing mapped reads to the transcriptome were converted to BedGraph files using Bedtools (v2.27.1) [90] and plotted using customized R scripts. To analyse the lengths of individual gene body regions (5' UTRs, CDSs and 3' UTRs), the Ensembl 92 annotation file was used. Read density by region was defined as number of reads per length of the region of interest. Number of reads per region was calculated using HTSeq (v0.9.1) [88] and feature type (-f) as 'five_prime_UTR', 'CDS', or 'three_prime_UTR'. If a read belonged to two distinct regions, it was counted for both of them. The ratio of the read densities (read density in *ski7* mutants versus read density in WT) was used to assess relative enrichments of reads over distinct gene body regions. If wild-type and *ski7* read densities were zero, the value was not considered for the calculation of the ratio. Codon adaptation index (CAI) of the coding sequences of annotated genes was calculated using a command line version of the CAIcal program (v1.4) [91], the zebrafish codon usage table (obtained from the codon usage database (https://www.kazusa.or.jp/codon/)) and standard genetic code (-g 1). The CAI was calculate per gene (-s y). As control, the CAI was calculated for a number of random sequences (-n) equal to the size of the group being compared (or 500 if the group was bigger than 1000 sequences) and with the number of codons (-l) equal to the average length of the CDS being analyzed. GO term enrichment analyses were performed for up- and down-regulated genes separately using topGO (v2.34.0) [92]. A reduction of terms for visualization was performed with the online tool REVIGO.

## TMT-MS

TMT-MS was performed in biological triplicates. For each sample, 200 embryos from individual parents were collected at 4 hours post-fertilization. Embryos were dechorionated (1 mg/ml pronase) and batch-deyolked following [93]. Samples were lysed by sonication in SDT buffer (4% SDS, 0.1 M DTT, 0.1 M Tris-HCl pH 7.5). For downstream normalization, the STD buffer was supplemented with 1 pmol of dCas9 protein and Lambda exonuclease protein per 10 µl buffer. Samples were diluted with 200 µl of 8 M urea, 0.1 M Tris-HCl, pH 8.5 and centrifuged at 14,000 g for 20 minutes using a 30 kDa mass-cutoff 0.5 ml Amicon centrifugal filter (Merck). For alkylation of cysteines, samples were incubated for 30 minutes in the dark after addition of 100 µl of 0.1 M IAA, 8 M urea, 0.1 M Tris-HCl pH 8.5 and then centrifuged at 14000 g for 20 minutes. Samples were washed with 100ul of 8 M urea, 0.1 M Tris-HCl pH 8.5 and 100 µl of 100 mM Hepes pH 7.6 by centrifugation at 14,000 g for 20 min (four wash steps in total). Tryptic digests were performed overnight after addition of 46 µl of 100 mM Hepes pH 7.6 and 4 µl of 1 µg/µl Trypsin Gold (Promega).

TMT Labelling: Samples were acidified with 10% TFA and cleaned up with 50 mg Sep-Pak C18 columns (Waters), freeze-dried overnight and then dissolved in 100 µl of 100 mM Hepes. Samples were run on a monolithic column for quantification to use not more than 100 µg per sample for TMT labeling. Samples were labelled with TMT according to manufacter's instructions (Thermo Fisher). Labelled samples were pooled and cleaned up via a 50 mg Sep-Pak C18 column (Waters) and separated into fractions using a SCX system (UltiMate system (Thermo Fisher) and TSKgel SP-25W column (Tosoh Bioscience; 5 µm particles, 1 mm ID × 300 mm), flow rate of 30 µl/min. For the separation, a ternary gradient was used: buffer A (5 mM phosphate buffer pH 2.7, 15% ACN), buffer B (5 mM phosphate buffer pH 2.7, 1 M NaCl, 15% ACN), and buffer C (5 mM phosphate buffer pH 6, 15% ACN). 60 fractions were collected and stored at -80˚. The fractions were analysed with LC-MS. The nano HPLC system used was an UltiMate 3000 HPLC RSLC nano system (Thermo Scientific) coupled to a Q Exactive HF-X mass spectrometer (Thermo Scientific), equipped with a Proxeon nanospray source (Thermo Scientific). Peptides were loaded onto a trap column (Thermo Scientific, PepMap C18, 5 mm × 300 µm ID, 5 µm particles, 100 Å) at a flow rate of 25 µL min$^{-1}$ using 0.1% TFA as mobile phase. After 10 min, the trap column was switched in line with the analytical column (Thermo Scientific, PepMap C18, 500 mm × 75 µm ID, 2 µm, 100 Å). A 180 min binary gradient from buffer A (water/formic acid, 99.9/0.1, v/v) to B (water/acetonitrile/formic acid, 19.92/80/0.08, v/v/v) was applied to elute the peptides at a flow rate of 230 nl min$^{-1}$. The Q Exactive HF-X mass spectrometer was operated in data-dependent mode, using a full scan (m/z range 375–1650, nominal resolution of 120,000, target value 3E6) followed by MS/MS scans of the 10 most abundant ions. MS/MS spectra were acquired using normalized collision energy of 35, isolation width of 0.7 m/z, resolution of 45.000, a target value of 1E5, and maximum fill time of 250 ms. For the detection of the TMT reporter ions, a fixed first mass of 110 m/z was set for the MS/MS scans. Precursor ions selected for fragmentation (exclude charge state 1, 7, 8, >8) were put on a dynamic exclusion list for 60 s. Additionally, the minimum AGC target was set to 1E4 and intensity threshold was calculated to be 4E4. The peptide match feature was set to preferred and the exclude isotopes feature was enabled.

For peptide identification, the RAW files were processed in Proteome Discoverer (v2.3.0.523, Thermo Scientific). All MS/MS spectra were searched using MSAmanda (v2.0.0.141114) [78] against a custom *Danio rerio* protein database (based on GRCz11 and annotated gene models from the Ensembl 92 release: 38,422 sequences, 21,882,367 residues supplemented with the sequences of the spike-ins). The parameters used were as follows: fixed modification: Iodoacetamide derivative on cysteine; variable modifications: deamidation on

asparagine and glutamine, oxidation on methionine, phosphorylation on serine, threonine and tyrosine, sixplex tandem mass tag on lysine, as well as carbamylation and simplex tandem mass tag on peptide N-termini. Trypsin was defined as the proteolytic enzyme allowing for up to two missed cleavages. Monoisotopic masses were searched within unrestricted protein masses for tryptic enzymatic specificity. The peptide mass tolerance was set to ± 5 ppm and the fragment mass tolerance to ± 15 ppm. The maximal number of missed cleavages was set to two. Identified spectra were FDR-filtered to 1.0% on peptide level using the Percolator algorithm integrated in Thermo Proteome Discoverer. The localization of the modification sites within the peptides was performed with ptmRS, which is based on phosphoRS [79]. Peptides were quantified based on Reporter Ion intensities as extracted by the "Reporter Ions Quantifier"-node as implemented in Proteome Discoverer. Proteins were quantified by summing up unique and razor peptides. Protein area normalization was done using the spiked-in proteins dCas9 and Lambda exonuclease. Missing values were imputed by the 5-percentile of the respective sample. Differentially expressed proteins were determined using limma [82] and p-values were adjusted for multiple testing using the limma R package [83].

## Reductive agent sensitivity test in Zebrafish embryos

To assess the sensitivity of wild-type and mutant embryos to DTT, we collected embryos from two to four pairs of fish from each genotype up to 20 minutes post-fertilization. Embryos were split into two groups. Group one (DTT treatment) was transferred to 0.2 mM DTT in blue water (E3 medium with 0.1% methylene blue) at 30 minutes post-fertilization. Embryos were kept in this medium for the duration of the experiment. Group two (untreated) consisted of untreated embryos (not incubated in DTT); untreated embryos were kept alongside as controls. Manual examination of defects was performed blindly at every hour for the first 8 hours post-fertilization.

## Statistical analyses

For the quantification of fertilization rates, Kruskal-Wallis with Dunn's multiple comparison test was performed, taking the cross of wild-type females and wild-type males as reference.

The significance of the proportion of overlapping genes within and among periods was calculated by Pearson's chi-squared test with Yates continuity correction, which tests for equality of proportions between two populations.

Significance of the differences in level of expression of DEGs and unchanged genes across the time course, as well as the read density over the gene body regions per time point were assessed by Wilcoxon test.

Significances of differences in transcript lengths, as well as for CAI comparisons were calculated by the Kolmogorov-Smirnov test.

To evaluate significant differences in viability under reductive conditions, Kruskal-Wallis with Dunn's multiple comparison test was performed per time-point, taking the wild-type untreated samples for every time point as reference.

Data visualization and all statistical analyses were performed using R (v3.5.1).

## Supporting information

**S1 Fig.** *Ski7* **expression in mouse and zebrafish.** (A) Expression levels (TPM) of mouse *Ski7* (purple) and *Hbs1l* (gray) during early embryogenesis. Left: *Hendrickson PG, et al 2017*; right: *Yu C, et al, 2016*. (B) Zebrafish Ski7 mutant protein lacks the conserved Ski7-like and P4-like motifs. Schematic (left) and amino acid sequence (right) of wild-type and mutant Ski7 proteins. Specific domains and motifs are highlighted. The mutant protein contains the shared N-

terminus from Hbs1l but lacks the Ski7-like and P4-like motifs due to a premature stop codon.
(C) *Hbs1l* mRNA levels remain unchanged in *ski7*$^{-/-}$ mutants. Screenshot of RNA-seq reads
over the *ski7* gene locus of mutant and wild-type samples. Red box indicates the 203-nt dele-
tion in the mutants (top). TPM levels of *ski7* (purple) and *hbs1l* (gray) in *WT* (straight line)
and *ski7*$^{-/-}$ (dotted line) across the entire time-course show no differences in expression for
*hbs1l.*
(TIF)

**S2 Fig. *Ski7* mutants are viable and develop into morphologically normal adults.** Represen-
tative images of wild-type and *ski7*$^{-/-}$ mutant female (top) and male (bottom) fish. No morpho-
logical differences are observed.
(TIF)

**S3 Fig. Ski7 mutant fish produce a higher proportion of eggs of poor quality and a reduced
number of developing offspring.** (A) Representative images and quantification of eggs of
good quality (GQ) and poor quality (PQ) at 30 minutes after egg-laying. (1) Good egg quality,
(2) egg without chorion elevation, and (3) opaque egg. Every bar represents an individual
female. Numbers on top of the bars indicate the number of eggs. (B) Percentage of developing
embryos from wild-type, mutant, and reciprocal crosses. P-adjusted from Kruskal-Wallis with
Dunn's comparison test (comparisons were performed against the cross of *WT* male with *WT*
female). *WT-WT* and *ski7*$^{-/-}$—*ski7*$^{-/-}$ crosses are from main Fig 1D.
(TIF)

**S4 Fig. Differential gene expression analysis in the absence of Ski7 based on rRNA-
depleted RNA-seq data.** (A) Number of up- and down-regulated genes in *ski7*$^{-/-}$ mutants dur-
ing oogenesis (O1 –O4), and embryogenesis (E1 –E4) based on rRNA-depleted mRNA enrich-
ment. (B) Venn diagrams of the number and percentage (in brackets) of DEGs obtained by
polyA$^{+}$, rRNA-, or both mRNA enrichment methods.
(TIF)

**S5 Fig. High correlation between polyA$^{+}$ and rRNA-based gene expression values of DEGs.**
Scatter plots of polyA$^{+}$ and rRNA- log fold changes of DEGs identified by polyA$^{+}$ RNA during
oogenesis and embryogenesis. Colored dots represent individual genes that were classified as
significantly differentially expressed in both methods. The number (bottom right corner) indi-
cates Spearman's ran correlation coefficient.
(TIF)

**S6 Fig. DEGs are shared within but not between periods.** (A) Number of up- and down-reg-
ulated genes in *ski7*$^{-/-}$ mutants during oogenesis (O), in mature eggs (M), and embryogenesis
(E). (B) Venn diagrams of differentially expressed genes for the three periods. Note that there
is almost no overlap of genes between all three periods.
(TIF)

**S7 Fig. Targets of Ski7 are broadly expressed during the oocyte-to-embryo transition.** (A)
Venn diagrams of the overlap of DEGs from the three periods from S6B Fig. (B) Venn dia-
grams of the overlap of 'period-specific' DEGs (differentially regulated in all stages of a given
period) during the oocyte-to-embryo transition in *WT* and *ski7*$^{-/-}$ samples. The numbers
below refer to the expression in wild type (left) or *ski7*$^{-/-}$ (right). Identified = expressed in wild
type or *ski7*$^{-/-}$ in at least one period; ≥2 periods = expressed in wild type or *ski7*$^{-/-}$ in at least
two periods; X + Y|Z = expressed in the period of origin (X) and at least one other period (Y
and/or Z).
(TIF)

**S8 Fig. Some of the Ski7's targets are regulated at a specific time.** (A) Venn diagrams of the overlap of DEGs from the three periods from S6B Fig. (B) Venn diagrams of the overlap of 'period-specific' DEGs (differentially regulated in all stages of a given period) compared to unchanged genes in the other periods. The numbers below refer to DEGs of the indicated period identified as unchanged in other periods.
(TIF)

**S9 Fig. Metagene profiles of DEGs and unchanged genes during the oocyte-to-embryo transition.** Metagene profiles of up-regulated, down-regulated, and unchanged genes of wild-type (gray) and mutants (purple) at all stages during the oocyte-to-embryo transition.
(TIF)

**S10 Fig. Down-regulated genes in the absence of Ski7 tend to be shorter during oogenesis.** Cumulative distribution of transcript lengths of up-regulated (orange), down-regulated (blue) and unchanged (gray) genes per period. (A) Analyses of the length of full transcripts shows that down-regulated genes during oogenesis tend to be shorter while up-regulated are longer (Kolmogorov-Smirnov $p = 1.7e^{-07}$ & 0.003, respectively). (B) Analyses of the length per transcript region (5' UTRs, CDSes and 3' UTRs). Note that the biggest difference in down-regulated genes during oogenesis is observed in CDSes and 3' UTRs (Kolmogorov-Smirnov $p = 1.6e^{-06}$ & $4.0e^{-05}$, respectively), and 3' UTRs in up-regulated genes (Kolmogorov-Smirnov $p = 0.021$).
(TIF)

**S11 Fig. Genes up-regulated in *ski7* mutants during oogenesis are enriched for rare codons.** Cumulative fraction of the usage of synonymous codons in DEGs (up-regulated: orange, down-regulated: blue) and unchanged genes (gray) per period as measured by codon adaptation index (CAI). P-values from Kolmogorov-Smirnov test.
(TIF)

**S12 Fig. Differentially expressed proteins in *ski7*^-/- embryos.** (A) Volcano plot of proteins identified by tandem mass tag mass spectrometry (TMT-MS) from wild-type and *ski7*^-/- mutant embryos at 4 hours post-fertilization. Significantly up- and down-regulated proteins are coloured in orange and blue, respectively. Factors involved in response to stress/oxidative stress are highlighted in green. Labelled dots highlight proteins for which the mRNA was also identified as differentially regulated by RNA-seq. In bold, oxidative stress proteins that were also identified by RNA-seq. (B) List of all significantly up- and down-regulate proteins (indicated by gene name) from the TMT-MS analyses. Differentially expressed proteins for which the mRNA was also identified as differentially expressed by RNA-seq are highlighted in bold. Genes that have been associated with stress response or oxidative/reductive stress are highlighted in green.
(TIF)

**S13 Fig. GO terms enriched in differentially expressed genes in *ski7* mutants.** GO terms enriched in the overlapping set of DEGs from each period (O = oogenesis, M = mature eggs, E = embryogenesis).
(TIF)

**S14 Fig. GO terms enriched in differentially expressed genes for each stage across the oocyte-to-embryo transition.** GO terms enriched in DEGs at each stage of the time-course

experiment during the oocyte-to-embryo transition. O = oogenesis, M = mature eggs,
E = embryogenesis.
(TIF)

**S1 Table. Conservation analysis of Ski7 protein homologs.** Species name, identifiers, and
databases of all Ski7 proteins used for the alignment in Fig 1C.
(XLSX)

**S2 Table. Quantification of poor-quality eggs.** Numbers of eggs of poor quality of individual
wild-type and *ski7* mutant females.
(XLSX)

**S3 Table. Percentage of developing embryos.** Numbers and percentages of developing
embryos of wild-type, *ski7* mutant, and rescued crosses.
(XLSX)

**S4 Table. Quantification and staging of developing embryos over time.** Scoring of embryos
over time up to 7 hours post-fertilization (hpf). (A-J) Individual experiments of wild-type
embryos. (I-P) Individual experiments of *ski7* mutant embryos.
(XLSX)

**S5 Table. Mass-spectrometric analysis of the Ski7-P4-like peptide pull-down.** Values of the
normalized abundances of detected proteins in mass-spectrometry of three independent repli-
cates of Ski7-P4-like peptide and a negative control. Log2 fold-change of peptide/negative con-
trol, as well as p-values are shown.
(XLSX)

**S6 Table. Overview of RNA-seq experiments.** (A) Number of reads per sample from 3 repli-
cates per genotype per stage; polyA$^+$ RNA-seq and ribo- (RiboC) RNA-seq). (B) Pearson corre-
lation between samples of polyA$^+$ RNA-seq. (C) Pearson correlation between samples of
RiboC RNA-seq.
(XLSX)

**S7 Table. Differentially expressed genes of polyA+ RNA-seq data during oogenesis.** List of
differentially expressed genes during oogenesis based on polyA+ RNA-seq. The list shows the
gene name (Gene), counts per million (cpm) per replicate for WT and *ski7*$^{-/-}$ mutants, the cal-
culated log2FC, and FDR (false discovery rate). The last column provides the classification of
genes as either significantly upregulated (UP) or downregulated (DOWN) (all other genes are
listed as FALSE) according to EdgeR.
(XLSX)

**S8 Table. Differentially expressed genes of polyA+ RNA-seq data in mature eggs.** List of
differentially expressed genes in eggs based on polyA+ RNA-seq. The list shows the gene name
(Gene), counts per million (cpm) per replicate for WT and *ski7*$^{-/-}$ mutants, the calculated
log2FC, and FDR (false discovery rate). The last column provides the classification of genes as
either significantly upregulated (UP) or downregulated (DOWN) (all other genes are listed as
FALSE) according to EdgeR.
(XLSX)

**S9 Table. Differentially expressed genes of polyA+ RNA-seq data during embryogenesis.**
List of differentially expressed genes during embryogenesis based on polyA+ RNA-seq. The
list shows the gene name (Gene), counts per million (cpm) per replicate for WT and *ski7*$^{-/-}$
mutants, the calculated log2FC, and FDR (false discovery rate). The last column provides the

classification of genes as either significantly upregulated (UP) or downregulated (DOWN) (all other genes are listed as FALSE) according to EdgeR.
(XLSX)

**S10 Table. Differentially expressed genes of Ribo- RNA-seq data during oogenesis.** List of differentially expressed genes during oogenesis based on Ribo- RNA-seq. The list shows the gene name (Gene), counts per million (cpm) per replicate for WT and *ski7*$^{-/-}$ mutants, the calculated log2FC, and FDR (false discovery rate). The last column provides the classification of genes as either significantly upregulated (UP) or downregulated (DOWN) (all other genes are listed as FALSE) according to EdgeR.
(XLSX)

**S11 Table. Differentially expressed genes of Ribo- RNA-seq data during embryogenesis.** List of differentially expressed genes during embryogensis based on Ribo- RNA-seq. The list shows the gene name (Gene), counts per million (cpm) per replicate for WT and *ski7*$^{-/-}$ mutants, the calculated log2FC, and FDR (false discovery rate). The last column provides the classification of genes as either significantly upregulated (UP) or downregulated (DOWN) (all other genes are listed as FALSE) according to EdgeR.
(XLSX)

**S12 Table. Binned values for metagene profiles.** Normalized values on 100 bins of the differentially expressed and unchanged genes, per stage, per replicate of wild-type and *ski7* mutant samples.
(XLSX)

**S13 Table. Transcript lengths of differentially expressed genes.** Length of the different gene body regions of the longest annotated transcript of every gene for differentially expressed and unchanged genes. Lengths of full transcript (A), 5' UTR (B), CDS (C), and 3' UTR (D) are indicated per period.
(XLSX)

**S14 Table. Codon adaptation index values.** Codon adaptation index values of up-regulated (A), down-regulated (B), and unchanged (C) genes on every period.
(XLSX)

**S15 Table. Mass-spectrometric analysis of TMT-MS of *WT* and *ski7*$^{-/-}$ embryos at 4 hours post fertilization.** Values of the raw and normalized abundances of proteins detected from TMT-MS of wild-type and *ski7*$^{-/-}$ 4hpf-embryos in triplicates. Log2 fold-change of *ski7*$^{-/-}$/wild-type and p-value (limma) are also displayed.
(XLSX)

**S16 Table. Quantification of DTT stress.** Scoring of untreated and 0.2 mM DTT treated wild-type and *ski7*$^{-/-}$ mutant embryos over time (1, 2, 3, 4, 6, 8 hours post-fertilization).
(XLSX)

## Acknowledgments

We thank the proteinchemistry facility at the ViennaBiocenter, particularly Richard Imre, Michael Schutzbier and Gerhard Dürnberger, for TMT-MS sample preparation and support in proteomics data analyses; Karin Aumayer and her team of the biooptics facility at the ViennaBiocenter, particularly Pawel Pasierbek, for support with microscopy; Mathias Madalinski for synthesizing the Ski7-P4-like peptide; the VBCF NGS Unit (www.viennabiocenter.org/facilities) for RNA sequencing; Maria Novatchkova, Dominik Handler, and Brian Reichholf

for helpful comments regarding RNA-seq data analyses; Karin Panser, Carina Pribitzer and the animal facility personnel for taking care of zebrafish; Mirjam Binner for help with genotyping; the VBC RNA Salon for helpful discussions and suggestions; the entire Pauli lab, particularly Anastasia Chugunova and Krista Gert for valuable discussions and comments on the project and manuscript.

## Author Contributions

**Conceptualization:** Luis Enrique Cabrera-Quio, Andrea Pauli.

**Data curation:** Luis Enrique Cabrera-Quio.

**Formal analysis:** Luis Enrique Cabrera-Quio.

**Funding acquisition:** Luis Enrique Cabrera-Quio, Karl Mechtler, Andrea Pauli.

**Investigation:** Luis Enrique Cabrera-Quio, Alexander Schleiffer, Andrea Pauli.

**Methodology:** Luis Enrique Cabrera-Quio, Karl Mechtler, Andrea Pauli.

**Project administration:** Karl Mechtler, Andrea Pauli.

**Resources:** Luis Enrique Cabrera-Quio, Alexander Schleiffer, Karl Mechtler.

**Software:** Luis Enrique Cabrera-Quio.

**Supervision:** Karl Mechtler, Andrea Pauli.

**Validation:** Luis Enrique Cabrera-Quio.

**Visualization:** Luis Enrique Cabrera-Quio.

**Writing – original draft:** Luis Enrique Cabrera-Quio, Andrea Pauli.

**Writing – review & editing:** Luis Enrique Cabrera-Quio, Andrea Pauli.

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
