## [Editor Report · Decision Letter 0]

9 Nov 2020

Dear Andi,

Thank you very much for submitting your Research Article entitled 'Zebrafish Ski7 tunes RNA levels during the oocyte-to-embryo transition' to PLOS Genetics. Since this was a manuscript transmitted from Review Commons, your manuscript was fully evaluated at the editorial level for its appropriateness at PLOS Genetics. The editors appreciated the strength of the transcriptomic analysis to examine the function of Ski7 in the oocyte to embryo transition in the zebrafish. The defect reported in fertility, however, requires further analysis prior to resubmission and review, as noted below. Additional more minor points below should also be addressed.

We request that analysis of fertilization itself be examined in the ski7 mutant, which is currently lacking. For example, does the sperm/male pronucleus enter the egg?  If it does enter, does pronuclear congression occur? If not, is the micropyle present or Bouncer affected?

Fertility decreases in older fish, so it is relevant to note if  WT and transgene females used were of the same age as the mutant females?  Were the same males used for the WT and mutant females and what were their ages?  Were WT and transgene females used in the fertilization assays siblings to the ski7 mutant females to ensure a similar genetic background in the mixed TLAB strain used.

Clarify in Fig 5C if each dot represents a different female.  If not, how many females were used and were the controls siblings of the same age? At least 3 different females of each strain should be used.

In Fig 1B and 2A, please provide more specific information on the x-axis for the stages examined.

Add in Methods how O3 and O2 oocytes were identified.

We therefore ask you to modify the manuscript according to these recommendations before we can consider your manuscript further. Your revisions should address the specific points made by the editors.

We hope to receive your revised manuscript within the next 60 days. If you anticipate any delay in its return, we would ask you to let us know the expected resubmission date by email to plosgenetics@plos.org.

You can use this link to log into the system when you are ready to submit a revised version, having first consulted our Submission Checklist.

[LINK]

Yours sincerely,

Mary

Mary C. Mullins

Associate Editor

PLOS Genetics

Gregory P. Copenhaver

Editor-in-Chief

PLOS Genetics

---

## [Decision Letter · Decision Letter 1]

18 Jan 2021

Dear Andi,

Thank you very much for submitting your Research Article entitled 'Zebrafish Ski7 tunes RNA levels during the oocyte-to-embryo transition' to PLOS Genetics.

The manuscript was fully evaluated at the editorial level and by independent peer reviewers. The reviewers are very positive about the revised manuscript but identified some minor concerns that we ask you to address in a revised manuscript. 

Additionally, the editors note that since fertility was not able to be addressed by assaying for fertilization, the sub-fertility defect is presumptive and not known. Other mutants in zebrafish are known to be fertilized but do not progress in development beyond the 1-cell stage, which could also be the case here. Such mutants do not affect fertility per se but instead are maternal-effect mutants.  Please clarify this point in the text and then it can be stated to be a presumptive subfertility defect.

We therefore ask you to modify the manuscript according to the review recommendations. Your revisions should address the specific points made by each reviewer.

[LINK]

All the best,

Mary

Mary C. Mullins

Associate Editor

PLOS Genetics

Gregory P. Copenhaver

Editor-in-Chief

PLOS Genetics

Reviewer's Responses to Questions

**Comments to the Authors:**

Reviewer #1: The authors have nicely addressed the concerns I raised in the previous review, and the manuscript is far stronger as a result. In particular, the addition of a transgenic rescue and the increased numbers and depth of the RNAseq analysis were very important clearly described.

However, I have one remaining concern with the text. I think it is misleading to describe the ski7 mutants as having two defects, reduced egg quality and subfertility. The authors provide no evidence that the subfertility is a function of anything other than a defect in egg quality. Eggs that do not appear “normal-looking” are clearly abnormal, but eggs that DO appear normal under a dissecting microscope may not be completely normal. Given that many eggs have reduced morphological quality it is likely that the normal appearing eggs are also impaired. The only way to truly answer this question would be to perform a timed degradation of ski7 mRNA or protein that did not impact ski7 levels in oocytes. Clearly this experiment would be beyond the scope of the current manuscript, but the text should be modified to accommodate this concern.

There are typos on lines 584 and 585.

Reviewer #2: This is a revised manuscript and the authors have addressed the previous critiques through additional experiments and edits to the text. The findings are not supported by the data and the manuscript is suitable for publication.

Reviewer #3: This is a really interesting manuscript describing the exosome-dependent remodeling of the maternal transcriptome in the female gamete. Specifically, by studying the function the adaptor factor Ski7 in zebrafish. This paper is highly significant in research on early vertebrate development as it is the first molecular and functional characterization of Ski7 in an animal system, which poses it as a new regulator of the oocyte to embryo transition. This work also provides clear background and directions for future studies.

The authors have obviously worked hard on this project and present a massive amount of interesting data in this study. Since first review, the clarity of the manuscript has improved a lot. The authors have put a great effort in addressing all the previous comments. The topic of post-transcriptional regulation during the oocyte to embryo transition is also very understudied in vertebrates and I applaud this group for ‘stepping up’ and attempting to tackle this topic. However, I feel there are a few comments that need to be addressed before publication.

Major comments:

1. Based on data presented in figure S3 the authors conclude that ‘bad eggs’, in figure (3), display an opaque phenotype. Are these eggs lysed? Can the authors examine this lysis phenotype? If yes, it needs to be examined in the unactivated, activated and collected from natural mating eggs. Also, it would be nice to examine whether the lysis phenotype is observed after scoring in the no clear cell and small chorion ones. This might help to classify ‘bad eggs’ as those associated with a gradually triggered lysis instead.

2. One conclusion raised by the authors is that physiological levels of redox signaling might be regulated by Ski7 during the oocyte to embryo transition. Given that transcriptomic and proteomic data is available for mutants, it would be relevant to assess whether Ski7 loss-of-function impacts in the expression of apoptosis-related genes likely involved in poor egg quality acquisition and reduced fertility. Also, and from the perspective of oocyte maturation, it would be important to assess whether ovarian steroidogenesis disruption would lead to up-regulated apoptosis by analyzing the expression of somatic genes (i.e cyp19a1a) promoting their synthesis in the mutant stage III (O3) and IV (O4) oocytes. This, assuming that some but not all follicle cells have been removed from oocytes after sorting. This might help to propose a model where the redox activity of apoptosis inducing factors could be impacting on egg quality and fertility, and lead the authors to hypothesize how Ski7 functions during oocyte maturation.

3. Where is the Ski7 factor expressed? Given that a transgenic line is available, it would be nice to examine the expression of the Ski7 protein in somatic cells and oocytes. This might be able to clarify which ovarian cell type is associated with the oocyte maturation defect in the ski7 mutant.

Minor comments:

1. The causes and molecular predictors of zebrafish egg quality have also been described by others (i.e Yilmaz et al., 2017 (PLoS ONE); Cheung et al., 2019 (BMC Genomics)). It would be great if the authors discuss these papers.

2. Line 100: P4 should be defined here.

3. Please change inactive to unactivated egg(s), and active to activated egg(s).

**Have all data underlying the figures and results presented in the manuscript been provided?**

Reviewer #1: Yes

Reviewer #2: Yes

Reviewer #3: Yes

PLOS authors have the option to publish the peer review history of their article (what does this mean?). If published, this will include your full peer review and any attached files.

Reviewer #1: No

Reviewer #2: No

Reviewer #3: No

---

## [Editor Report · Decision Letter 2]

1 Feb 2021

Dear Andi,

We are pleased to inform you that your manuscript entitled "Zebrafish Ski7 tunes RNA levels during the oocyte-to-embryo transition" has been editorially accepted for publication in PLOS Genetics. Congratulations!

All the best,

Mary

Mary C. Mullins

Associate Editor

PLOS Genetics

Gregory P. Copenhaver

Editor-in-Chief

PLOS Genetics

Comments from the reviewers (if applicable):

**Data Deposition**

http://datadryad.org/submit?journalID=pgenetics&manu=PGENETICS-D-20-01474R2

**Press Queries**

---

## [Editor Report · Acceptance letter]

13 Feb 2021

PGENETICS-D-20-01474R2 

Zebrafish Ski7 tunes RNA levels during the oocyte-to-embryo transition 

Dear Dr Pauli, 

We are pleased to inform you that your manuscript entitled "Zebrafish Ski7 tunes RNA levels during the oocyte-to-embryo transition" has been formally accepted for publication in PLOS Genetics! Your manuscript is now with our production department and you will be notified of the publication date in due course.

With kind regards,

Alice Ellingham

PLOS Genetics

On behalf of:
